# Accumulation of unacetylatable Snf2p at the *INO1* promoter is detrimental to remodeler recycling supply for *CUP1* induction

**Michelle Esposito**[1,2¤a], **Goldie Libby Sherr**[1,2¤b], **Anthony Esposito**[3], **George Kaluski**[1], **Farris Ellington**[1], **Chang-Hui Shen**[1,2,4,5]*

**1** Department of Biology, College of Staten Island, City University of New York, Staten Island, New York, United States of America, **2** PhD Program in Biology, The Graduate Center, City University of New York, New York, New York, United States of America, **3** Department of Biology, New Jersey City University, Jersey City, New Jersey, United States of America, **4** PhD Program in Biochemistry, The Graduate Center, City University of New York, New York, New York, United States of America, **5** Institute for Macromolecular Assemblies, City University of New York, Staten Island, New York, United States of America

¤a Current address: Department of Biology, Georgian Court University, Lakewood, New Jersey, United States of America
¤b Current address: Department of Biological Sciences, Bronx Community College, City University of New York, Bronx, New York, United States of America
* changhui.shen@csi.cuny.edu

**Data Availability Statement:** All relevant data are within the manuscript and its Supporting Information files.

## Abstract

Chromatin structure plays a decisive role in gene regulation through the actions of transcriptional activators, coactivators, and epigenetic machinery. These *trans*-acting factors contribute to gene expression through their interactions with chromatin structure. In yeast *INO1* activation, transcriptional activators and coactivators have been defined through intense study but the mechanistic links within these *trans*-acting factors and their functional implications are not yet fully understood. In this study, we examined the crosstalk within transcriptional coactivators with regard to the implications of Snf2p acetylation during *INO1* activation. Through various biochemical analysis, we demonstrated that both Snf2p and Ino80p chromatin remodelers accumulate at the *INO1* promoter in the absence of Snf2p acetylation during induction. Furthermore, nucleosome density and histone acetylation patterns remained unaffected by Snf2p acetylation status. We also showed that cells experience increased sensitivity to copper toxicity when remodelers accumulate at the *INO1* promoter due to the decreased *CUP1* expression. Therefore, our data provide evidence for crosstalk within transcriptional co-activators during *INO1* activation. In light of these findings, we propose a model in which acetylation-driven chromatin remodeler recycling allows for efficient regulation of genes that are dependent upon limited co-activators.

## Introduction

Transcriptional activation is a highly-regulated process that involves the recruitment of various transcription activators and coactivators to a gene's upstream regulatory region [1–2]. One critical step during the activation process is the participation of chromatin remodeling activities [3]. There are at least two main epigenetic mechanisms by which remodeling activities can

**Funding:** We would like to declare that CHS received funding from North Atlantic Treaty Organization (NATO) SPS G5266, National Science Foundation (NSF) MCB 0919218, Professional Staff Congress-City University of New York (PSC-CUNY) awards for this research. The funders had no role in study design, data collection and analysis, decision to publish, or preparation of the manuscript.

**Competing interests:** No authors have competing interests.

influence gene activation. The first is the recruitment of histone modifying enzymes to regulatory regions, which can perform various modifications, such as phosphorylation or acetylation, on histones [4]. The second is the recruitment of chromatin remodelers to perform nucleosome repositioning [5]. As such, biological functions of these remodeling activities have been defined.

*INO1* encodes for inositol-3-phosphate-synthase (I-3-P synthase), which converts glucose-6-phosphate (G-6-P) to inositol-3-phosphate (I-3-P). I-3-P is then dephosphorylated to form inositol by inositol monophosphatases, encoded by *INM1* and *INM2*. This inositol, which is repressive to *INO1*, then leads to the synthesis of PI through the actions of *PIS1*, which codes for PI synthase [6–7]. As such, *INO1* expression is responsible for the rate-limiting step of *de novo* phospholipid synthesis in yeast, and it has been studied intensively [8–11].

It has been demonstrated that the expression of *INO1* is an Ino2p activator-dependent event [12–13]. Ino2p binds as an Ino2p/Ino4p heterodimer to the *INO1* upstream activating sequence (*UAS*), and subsequently recruits coactivators, such as chromatin remodelers, Snf2p and Ino80p, as well as, histone acetylases, Esa1p and Gcn5p [8–11]. Chromatin remodelers, Snf2p and Ino80p, exhibit high occupancy levels before *INO1* induction, and then dissociate from the promoter as INO1 induction commences [10]. Histone acetyltransferases (HATs), Gcn5p and Esa1p, on the other hand, exhibit the opposite pattern of recruitment in which they are present in low numbers at the *INO1* promoter during repression, but greatly increase upon INO1 induction [11]. Therefore, the players that are involved in *INO1* activation have been identified.

Although both Snf2p and Ino80p are key remodelers of *INO1* expression [8–9], the interplay of these remodelers with other transcriptional coactivators present at the *INO1* promoter has not been fully elucidated. Recently, it has been suggested that chromatin remodelers can be regulated by post-translational modifications to their domains or various subunits. These modifications can include phosphorylation of SWI/SNF or acetylation of SWI/SNF, RSC, and SWI families [14–19]. Such modifications suggest mechanistic links between HAT and chromatin remodelers and this may have an impact on the regulation of gene expression. For example, it has been shown that HAT Gcn5p is capable of acetylating chromatin remodeler Snf2p, and this can lead to the dissociation of the SWI/SNF complex from acetylated histones, and reduces its association with promoters *in vivo* [17]. The implications of this acetylation and subsequent dissociation, however, have yet to be explored with regard to transcriptional regulation and other possible mechanisms.

In this study, we have performed biochemical analyses to examine the implications of Snf2p acetylation during *INO1* activation. We also examined whether or not Snf2p acetylation and occupancy changes influenced other gene activities, as many coactivators are involved in regulating large numbers of genes within cells and may need highly regulated recycling patterns. Our results showed that the accumulation of unacetylatable Snf2p at the *INO1* promoter may have significantly impacted other Snf2p-regulated genes' expression.

## Results

### Unacetylatable Snf2p leads to the accumulation of chromatin remodelers at the *INO1* promoter

Transcriptional activation is a highly regulated process that requires the recruitment of an assortment of activators and coactivators to a gene's upstream regulatory region. Previous studies have suggested that the post-translational modification of the chromatin remodeler Snf2p can lead to its dissociation from gene promoters [17]; however, this has yet to be studied for its biological implication on gene expression. To this end, we have chosen the *INO1* system to understand the implications of Snf2p acetylation at the *INO1* promoter. It has been shown

that both chromatin remodelers Ino80p and Snf2p depart from *INO1* promoter region once transcription is commenced [8, 11, 20]. Furthermore, both Esa1p and Gcn5p HATs were recruited right before the remodelers' departure. These results suggest that the action of departure might result from the remodelers' acetylation. To confirm the acetylation was indeed necessary for Snf2p dissociation, chromatin immunoprecipitation (ChIP) was performed using WT cells and an unAcSnf2p mutant in which the acetylatable lysine residues of the remodeler were replaced with unacetylatable arginine residues [17]. In both strains, it has been engineered so that Snf2p contained a C-terminal double-FLAG tag. A *snf2Δ* strain was utilized as a negative control.

For repressing and inducing conditions, the IP signals of Snf2p-FLAG were examined at the upstream regulatory sequence (*URS*) of *INO1* and were subsequently normalized to the *INO1* input and mock. In the WT strain, the IP values demonstrated a significant difference ($p<0.01$), as the Snf2p-FLAG-IP were 1.26% ±0.19% under repressing conditions and 0.10% ±0.01% under inducing conditions, which demonstrated the dissociation of Snf2p from the promoter region (Fig 1A). This result is in agreement with previous findings [10]. On the other hand, IP values for the Snf2p-FLAG-IP in the unAcSnf2p mutant were 1.13% ±0.17% and 1.12% ±0.13% under repressing and inducing conditions, respectively (Fig 1A). The difference is insignificant ($p = 0.996$), and this observation indicated that unacetylated Snf2p does not depart from the *INO1* promoter. The negative control, *snf2Δ*, demonstrated minimal IP values with no significant difference between repressing and inducing conditions, with 0.01% ±0.001% and 0.01% ±0.002% under repressing and inducing conditions, respectively. Therefore, our data demonstrate the accumulation of unacetylated Snf2p at the *INO1* promoter upon *INO1* induction in the absence of Snf2p acetylation, suggesting that Snf2p acetylation is critical to promoter dissociation.

Previously, it has been shown that both remodelers, Ino80p and Snf2p, are required at the *INO1* promoter to activate *INO1* transcription [10–11], and that both remodelers accumulate in the absence of the histone acetyltransferases, Esa1p and Gcn5p [11]. Since Ino80p is required to recruit Snf2p to the *INO1* promoter [10], it is interesting to examine how Snf2p accumulation can affect Ino80p at the *INO1* promoter in the absence of Snf2p acetylation. To this end, ChIP coupled with qPCR was performed on WT, unAcSnf2p, and *snf2Δ* strains.

IP signals using an antibody against Arp8p (Arp8p-IP) of INO80 were examined at the *INO1* URS under repressing and inducing conditions, respectively, and were subsequently normalized to *INO1* input and *INO1* mock DNA. In the WT strain, the IP values for the Arp8p-IP were 0.78±0.07 under repressing conditions and 0.17% ±0.03% upon induction of *INO1*, suggesting a significant dissociation of the Ino80p from the promoter upon *INO1* induction ($p<0.01$) (Fig 1B). Similarly, the Arp8p-IP were 0.74% ±0.13% with repressing conditions and 0.15% ±0.04% under inducing conditions in the *snf2Δ* strain as Ino80p arrives at the *INO1* prior to and independent of Snf2p [10]. In the unAcSnf2p mutant strain, on the other hand, the IP values for the Arp8p-IP in the repressed condition were statistically similar to the induced condition, 0.67% ±0.11% under repressing conditions and 0.61% ±0.12% under inducing conditions (Fig 1B). These results suggest the absence of Ino80p dissociation from the *INO1* promoter when Snf2p cannot be acetylated. Therefore, both Snf2p and Ino80p accumulate at the *INO1* promoter in the absence of Snf2p acetylation.

## The acetylation of Snf2p is not required for *INO1* activity

Since it was observed that Snf2p acetylation is required for Snf2p dissociation from the *INO1* promoter, it was subsequently instructed to determine the implication of acetylation on *INO1* expression through growth analysis. Since *INO1* plays a rate-limiting role in the *de novo*

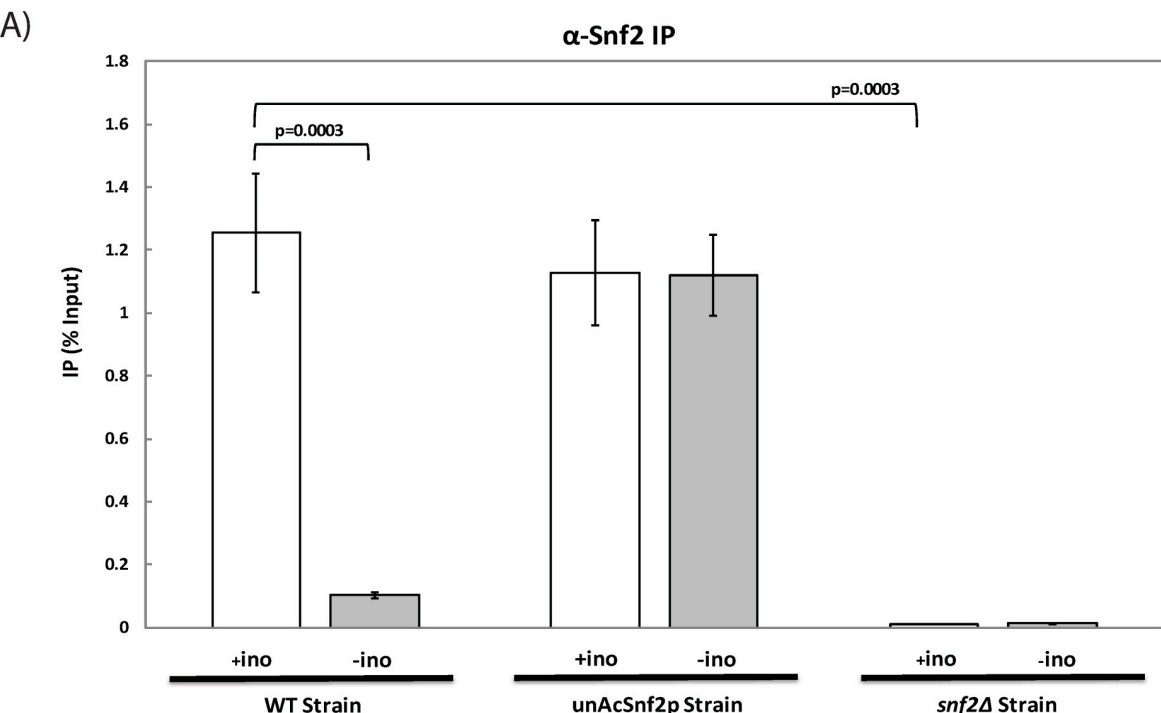

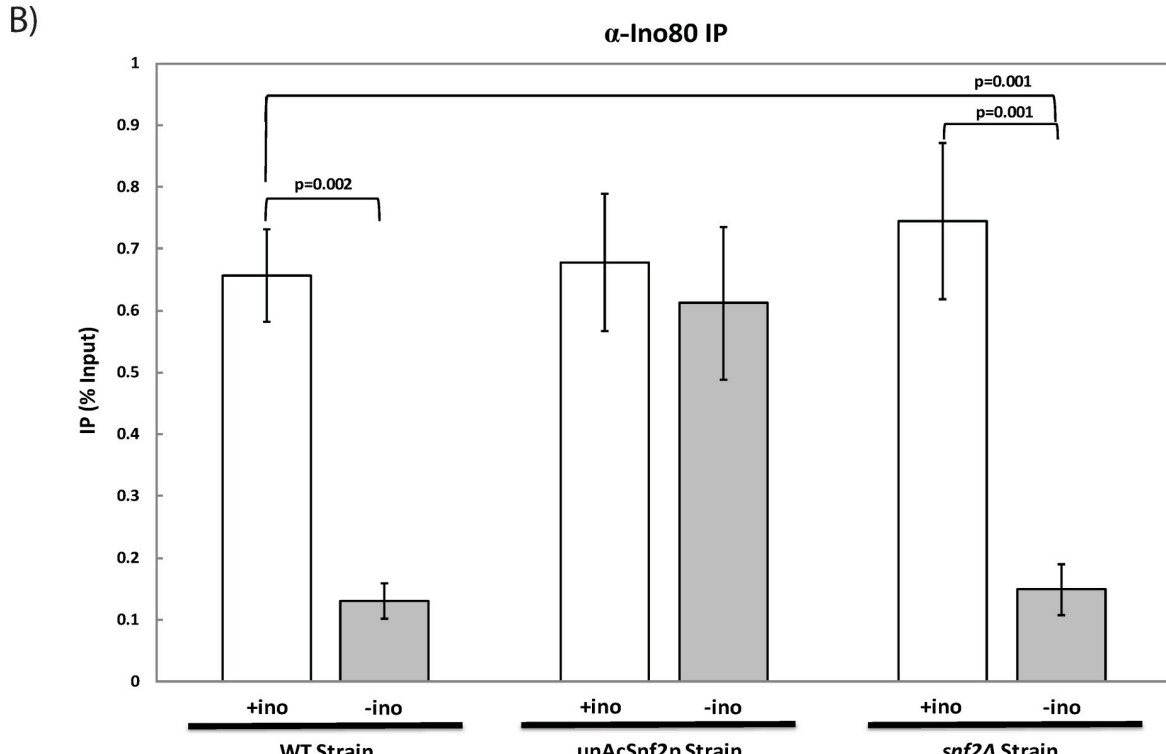

**Fig 1. Unacetylatable Snf2p leads to the accumulation of chromatin remodelers at the *INO1* promoter instead of dissociating.** Real-time PCR analysis of DNA immunoprecipitated through Chromatin Immunoprecipitation (ChIP) with an antibody against (A) FLAG-tagged Snf2p (α-Snf2p) or (B) Ino80p (α-Ino80p) in WT cells, unAcSnf2p cells, and *snf2Δ* cells that were grown to mid-log phase ($A_{600}$ = 1.0) in 10μM inositol Synthetic Complete (SC) media and were subsequently subjected to repressing or inducing conditions, 100 μM inositol (+ino; open bars) and 0 μM inositol (-ino; light grey bars) containing synthetic complete media, respectively, for 2 hours prior to collection. All experiments were performed with three repeats and all PCR reactions were done in at least triplicate. The IP for the *INO1* promoter is graphed as mean ± standard deviation normalized to input and mock.

synthesis of inositol in cells, cells that have impeded *INO1* expression are expected to demonstrate a growth deficiency in media lacking inositol, whereas cells with fully functioning *INO1* gene activity can successfully synthesize inositol that is absent from the environment and can thus still thrive in inositol depleted media.

In media containing 100 μM inositol, repressing conditions, the WT type strain thrived and reached an average peak optical density $A_{600nm}$ of approximately 7.9±0.07. In inositol depleted media, inducing conditions, the WT cells were still able to survive, but only reached an average peak $A_{600nm}$ of approximately 6.0±0.07 (Fig 2A). The generation times are 274.29 min/generation and 292.68 min/generation for the WT under repressing and inducing conditions,

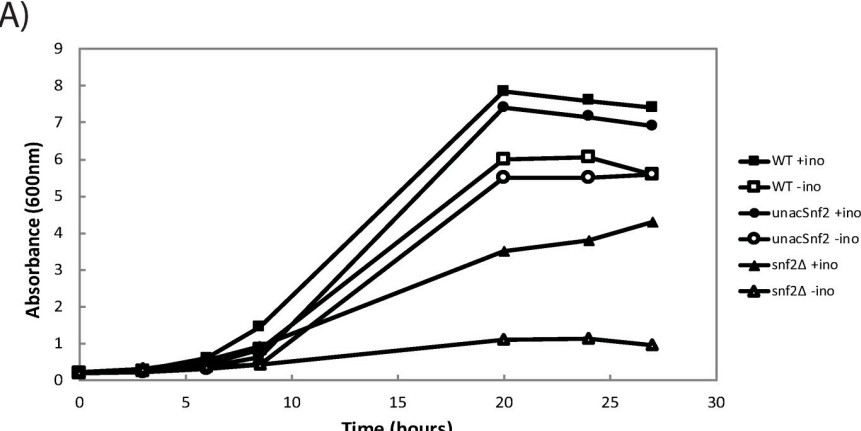

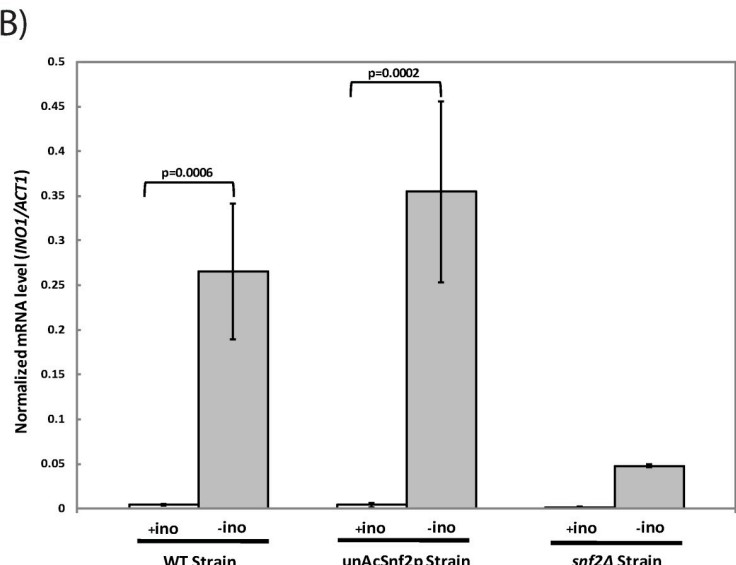

**Fig 2. The *INO1* induction is indepedent of Snf2p acetylation.** (A) Growth of yeast cells in the presence or absence of 100 μM myo-inositol. WT cell: (■) SC + inositol, (□) SC—inositol. Mutant UnAcSnf2p cell: (●) SC + inositol, (○) SC —inositol. Mutant *snf2Δ* cell: (▲) SC + inositol, (Δ) SC—inositol. Cells were grown in SC media and were subsequently washed and diluted to an $A_{600nm}$ of 0.2 in repressing or inducing conditions, 100μM inositol (+ino) and 0μM inositol (-ino) synthetic complete media, respectively. (B) *INO1* mRNA was examined by qRT-PCR analysis. Cells were grown in SC media to mid-log phase ($A_{600}$ = 1.0) and were subsequently washed and resuspended in repressing (+ino; open bars) or inducing (-ino; light grey bars) conditions, respectively. Cells were grown for 2 hours prior to collection. The expression ratio for *INO1* mRNA/*ACT11* mRNA is graphed as mean standard ± deviation. All experiments were repeated at least three times, and in each experiment PCR reactions were done in triplicate.

respectively. The unAcSnf2p strain demonstrated a growth pattern similar to the WT in both the *INO1* repressing and inducing conditions ($p$ = 0.3 and $p$ = 0.14, respectively), with an average peak $A_{600nm}$ around 7.5±0.14 in the presence of inositol and around 5.6±0.00 in the absence of inositol. Here, the generation times are 279.07 and 301.36 min/generation for the unAcSnf2p strain under repressing and inducing conditions, respectively. The *snf2Δ* strain, however, demonstrated a significantly different growth pattern compared to the unAcSnf2p strain ($p$<0.01 for repressing and inducing conditions), as the average peak $A_{600nm}$ in 100μM inositol was around 5.0±0.07, but in 0μM inositol it was only around 1.3±0.07. Taken together, these results demonstrated that the unAcSnf2p strain exhibited growth patterns very similar to the WT strain rather than the *snf2Δ* strain, suggesting that Snf2p acetylation is not required for cell growth in the absence of inositol. This implies that *INO1* expression is not influenced by the unacetylatable Snf2p.

To further identify that *INO1* expression is independent of the Snf2p acetylation, qRT-PCR analysis was employed. A significant difference in *INO1* mRNA levels between repressing and inducing conditions, 0.005±0.001 and 0.266±0.076, under repressing and inducing conditions, respectively; p<0.01, was observed in the WT strain, as well as in the unAcSnf2p strain, 0.004 ±0.003 and 0.355±0.101, under repressing and inducing conditions, respectively; $p$<0.01. On the other hand, no significant differences of *INO1* expression were observed in the *snf2Δ* strain (Fig 2B). This demonstrated that the *INO1* expression patterns in the unacetylatable strain were comparable to those observed in the WT strain rather than the deletion strain, as both strains demonstrated significant upregulation of *INO1* upon induction. Taken together with the growth analyses, the mRNA analyses confirmed that the acetylation of Snf2p was not critical for *INO1* activation.

## Nucleosome density, histone acetylation patterns and transcription machinery at *INO1* promoter are not influenced by the unacetylation of Snf2p

Since remodelers Snf2p and Ino80p accumulated at the *INO1* promoter and *INO1* expression is not influenced by the Snf2p unacetylation, it is interesting to see if remodeling activity is affected under the same condition. In order to examine nucleosome density, ChIP coupled with qPCR was performed in which antibodies targeted histones H3 and H4. In the WT strain, the H3-IP values demonstrated a significant difference in nucleosome density upon induction of *INO1* ($p$<0.01), which is expected as chromatin remodelers are modifying the chromatin structure. Under these same conditions, the unAcSnf2p mutant demonstrated a similar nucleosome density pattern as the wild type strain, with a significant difference in H3 nucleosome density upon *INO1* induction ($p$<0.01), which suggested that the chromatin remodeling activities at the *INO1* promoter were not affected by Snf2p unacetylation (Fig 3A). Similar results were also observed with regard to histone H4 ($p$ = 0.01; Fig 3B). These results suggest that chromatin structure is still subject to remodeling activities in the absence of Snf2p acetylation.

To evaluate histone acetylation, ChIP coupled with qPCR was performed using antibodies against acetylated H3 (acH3) and acetylated H4 (acH4). In the WT strain, the acH3-IP values were 0.11% ±0.005% and 1.22% ±0.26% under repressing and inducing conditions, respectively (Fig 3C). This demonstrated a significant increase in acetylated H3 upon induction ($p$<0.01). A similar pattern was also observed for the unAcSnf2p mutant in which the acH3-IP values were 0.07% ±0.01% and 0.90% ±0.04% under repressing and inducing conditions, respectively. On the contrary, the *snf2Δ* strain demonstrated no significant difference between repressing and inducing conditions (Fig 3C). Acetylated H4 also demonstrated a similar pattern in both the WT strain and the unAcSnf2p strain. Under WT repressing and inducing

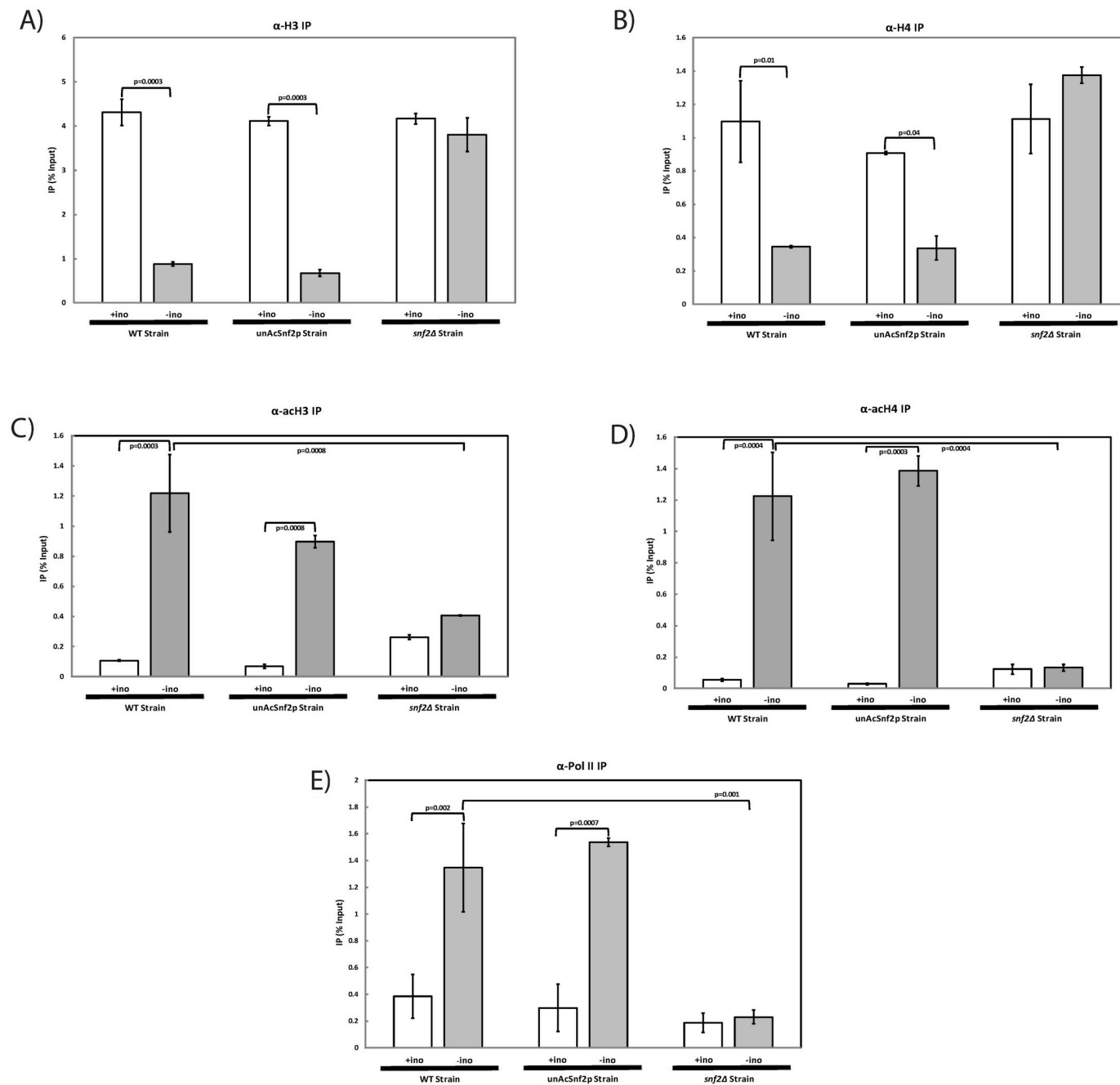

**Fig 3. Chromatin remodeling activities and the recruitment of transcription machinery at *INO1* promoter are independent of Snf2p acetylation.** Real-time PCR analysis of DNA immunoprecipitated through ChIP with an antibody against (A) Histone H3 (α-H3); (B) Histone H4 (α-H4); (C) Acetylated Histone H3 (α-acH3); (D) Acetylated Histone H4 (α-acH4); (E) RNA polymerase II (α-Pol II) at the *INO1* promoter for WT cells, unAcSnf2p, and *snf2Δ* cells that were grown to mid-log phase ($A_{600} = 1.0$) in SC and were subsequently washed and subjected to repressing (+ino; open bars) or inducing (-ino; light grey bars) conditions, respectively, for 2 hours prior to collection. All experiments were performed with three repeats and all PCR reactions were done in at least triplicate. The IP for the *INO1* promoter is graphed as mean ± standard deviation normalized to input and mock.

conditions, the acH4-IP values were 0.06% ±0.01% and 1.22% ±0.28%, respectively, which demonstrated a significant increase of H4 acetylation during induction conditions. In the unAcSnf2p mutant strain, a significant increase upon induction was also observed with acH4-IP values of 0.03% ±0.004% under repressing conditions and 1.39% ±0.10% under

inducing conditions, whereas the *snf2Δ* demonstrated no significant difference (Fig 3D). These results suggest that even though chromatin remodelers are accumulated at the *INO1* promoter in unAcSnf2p cells, the acetylation of histones in this region is still maintained.

We next wanted to examine RNA polymerase recruitment to the *INO1* promoter through IP using an antibody directed against RNA Pol II (Pol II). In the WT strain, Pol II-IP had IP values of 0.39% ±0.16% and 1.35% ±0.33% under repressing and inducing conditions, respectively. This showed a significant increase of RNA pol II recruitment during induction ($p<0.01$) (Fig 3E). A similar pattern was observed in the unAcSnf2p mutant strain, with IP values of 0.30% ±0.18% and 1.54% ±0.03% under repressing and inducing conditions, respectively, whereas the *snf2Δ* mutant exhibited no significant difference between repressing and inducing conditions. Taken together, these results confirmed that the recruitment of the RNA polymerase II to the *INO1* promoter was not dependent upon the acetylation of Snf2p. Furthermore, this validates the *INO1* expression results that *INO1* was expressed in unAcSnf2p mutant cells under inducing conditions (Fig 2B).

### *INO1* expression, nucleosome density, recruitment of remodelers, and remodeling activities are independent of Snf2p acetylation when switching from inducing conditions to repressing conditions

Since it was observed that the loss of Snf2p acetylation primarily affects the release of chromatin remodelers and that *INO1* expression is not influenced in the absence of Snf2p acetylation, we are interested in examining the impact if *INO1* is induced first, then subsequently exposed to repressing conditions. From the qRT-PCR analysis, *INO1* mRNA levels were high under inducing conditions for both WT strain and the unAcSnf2p strain (Fig 4A). However, *INO1* expression was downregulated for both WT and unAcSnf2p cells if the inducing conditions was employed for 30 min or 1 hr followed by the introduction of 1hr 30 min or 1 hr repressing conditions, respectively. This demonstrated that the *INO1* expression patterns in the unacetylatable strain were comparable to those observed in the WT strain, as both strains demonstrated significant upregulation of *INO1* upon induction and significant downregulation of *INO1* upon repression. Taken together, the mRNA analyses showed that the acetylation of Snf2p was not critical for *INO1* expression upon switching between *INO1* induction and repression switch.

Next, we wanted to examine the recruitment patterns of Snf2p and Ino80p. For inducing conditions, the IP signal of Snf2p-FLAG of WT strain was very low (Fig 4B). Upon the introduction of the repressing conditions after 1 hr induction, the IP values demonstrated a significant increase from 0.10%±0.002% to 1.26% ±0.09% ($p = 0.006$). On the other hand, the IP value for the Snf2p-FLAG-IP in the unAcSnf2p mutant was 1.12% ±0.14% under inducing conditions (Fig 4B). Upon switching the inducing conditions to repressing conditions, the IP value was 1.13% ±0.28%. The difference is insignificant, and this observation indicated that unacetylated Snf2p does not depart from the *INO1* promoter under inducing conditions. Upon switching the conditions to repressing conditions, large amounts of unacetylated Snf2p were at the *INO1* promoter. Since Ino80p also accumulated at *INO1* promoter when Snf2p was unacetylated (Fig 1B), we also examined the presence of Ino80p during the switch process. In the WT strain, the IP values for the Arp8p-IP were 0.13%±0.01% under inducing conditions and 0.66% ±0.003% upon switching to repression of *INO1*, suggesting a significant dissociation of the Ino80p from the promoter upon *INO1* induction and accumulation of Ino80p upon repression ($p = 0004$) (Fig 4C). In the unAcSnf2p mutant strain, on the other hand, the IP value for the Arp8p-IP in the inducing conditions was 0.61% ±0.06% (Fig 4C). Upon switching the inducing conditions to repressing conditions, the IP value was 0.68% ±0.01%. The

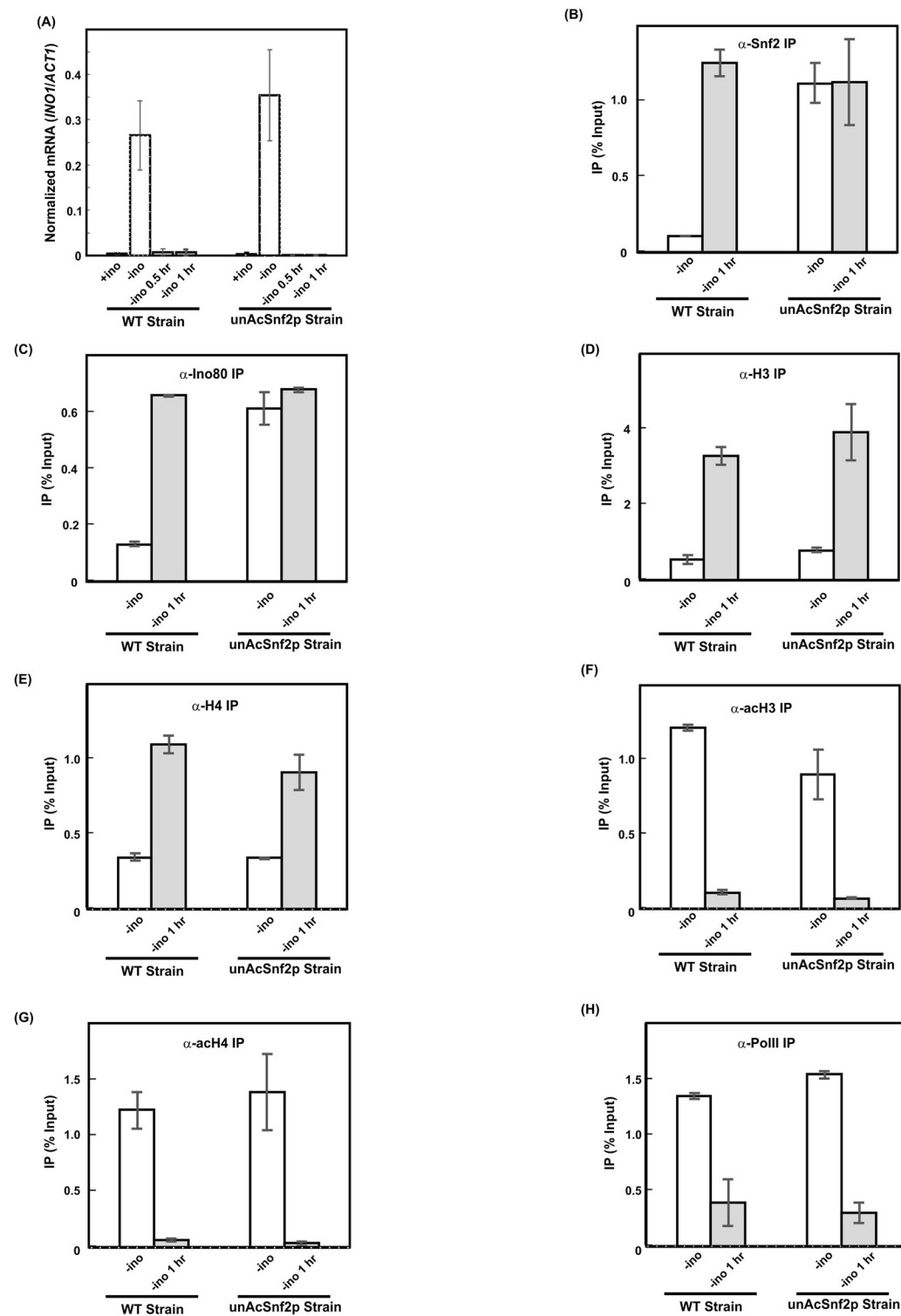

**Fig 4. The *INO1* induction and chromatin remodeling activities are indepedent of Snf2p acetylation when switching from inducing conditions to repressing conditions.** (A) *INO1* mRNA was examined by qRT-PCR analysis. Cells were grown in SC media to mid-log phase ($A_{600}$ = 1.0) and were subsequently washed and resuspended in inducing conditions for 30 min or 1hr, respectively. Subsequently, cells were switched to repressing conditions for another 1 hr 30 min or 1 hr prior to collection. The expression ratio for *INO1* mRNA/*ACT11* mRNA is graphed as mean standard ± deviation. All experiments were repeated at least three times, and in each experiment PCR reactions were done in triplicate. (B-H) Real-

time PCR analysis of DNA immunoprecipitated through Chromatin Immunoprecipitation (ChIP) with an antibody against (B) FLAG-tagged Snf2p (α-Snf2p), (C) Ino80p (α-Ino80p), (D) Histone H3 (α-H3); (E) Histone H4 (α-H4); (F) Acetylated Histone H3 (α-acH3); (G) Acetylated Histone H4 (α-acH4); (H) RNA polymerase II (α-Pol II) at the *INO1* promoter for WT cells and unAcSnf2p cells that were grown to mid-log phase ($A_{600}$ = 1.0) in SC and were subsequently washed and subjected to inducing (-ino; open bars) or switching (1 hr inducing followed by 1 hr repressing conditions) (-ino 1hr; light grey bars) conditions, respectively, for total 2 hours incubation prior to collection. All experiments were performed with three repeats and all PCR reactions were done in at least twice. The IP for the *INO1* promoter is graphed as mean ± standard deviation normalized to input and mock.

difference is insignificant. Our results demonstrated that both Snf2p and Ino80p accumulate at the *INO1* promoter in the absence of Snf2p acetylation even if we introduced inducing conditions followed by the repressing conditions.

For nucleosome density, the H3-IP value of WT strain demonstrated very low nucleosome density which is 0.52%±0.12% upon induction of *INO1* (Fig 4D). Upon switching the inducing conditions to repressing conditions, the H3-IP value increased to 3.29%±0.25% ($p$ = 0.01). Under these same conditions, the unAcSnf2p mutant demonstrated a similar nucleosome density pattern (from 0.79%±0.06% to 3.91%±0.74%; $p$ = 0.05) as the wild type strain, which suggested that the chromatin remodeling activities at the *INO1* promoter were not affected by Snf2p unacetylation. Similar results were also observed with regard to histone H4 (Fig 4E). Therefore, chromatin structure is subject to remodeling activities and is independent of Snf2p acetylation. Furthermore, we found that the *INO1* chromatin structure is independent of the initial conditions. When the conditions were switched from induction to repression, the chromatin structures of both WT and unAcSnf2p strains were similar to the chromatin structure at the repressing conditions (Figs 3A, 3B, 4D and 4E).

For the histone acetylation, the acH3-IP value of the WT strain was 1.22% ±0.02% under inducing conditions (Fig 4F). Upon switching the conditions from induction to repression, the acH3-IP value was 0.11% ±0.01% ($p$ = 0.0005). This demonstrated a significant decrease in acetylated H3 when the condition was switched to repression. A similar pattern was also observed for the unAcSnf2p mutant in which the acH3-IP values were 0.90% ±0.17% and 0.07% ±0.01% under inducing and switching to repressing conditions, respectively ($p$ = 0.04) (Fig 4F). Acetylated H4 also demonstrated a similar pattern in both the WT strain and the unAcSnf2p strain. Under WT inducing and switching to repressing conditions, the acH4-IP values were 1.22% ±0.17% and 0.06% ±0.01% ($p$ = 0.02), respectively, which demonstrated a significant decrease of H4 acetylation when switched to repressing conditions. In the unAcSnf2p mutant strain, a significant decrease upon switching to repressing conditions was also observed with acH4-IP values of 1.39% ±0.34% under inducing conditions and 0.03% ±0.016% when switched to repressing conditions ($p$ = 0.05) (Fig 4G). These results suggest that even though chromatin remodelers are accumulated at the *INO1* promoter in the unAcSnf2p strain, histone acetylation in this region is similar to WT strain.

We next examined RNA polymerase recruitment to the *INO1* promoter. In the WT strain, Pol II-IP had IP values of 1.35% ±0.02% and 0.39% ±0.21% under inducing and switching from inducing to repressing conditions, respectively ($p$ = 0.05) (Fig 4H). A similar pattern was observed in the unAcSnf2p mutant strain, with IP values of 1.54% ±0.03% and 0.30% ±0.10% under inducing conditions and switching to repressing conditions, respectively ($p$ = 0.007). These results confirmed that the recruitment of the RNA polymerase II to the *INO1* promoter was also independent of the acetylation of Snf2p. Taken together, our results showed that the recruitment of chromatin remodelers, chromatin structure, remodeling activities and the transcription machinery are all similar between WT strain and unAcSnf2p strain when switched from inducing to repressing conditions, suggesting that they all are independent of Snf2p acetylation during the transition period.

## Increased copper sensitivity when Ino80p and Snf2p accumulate at the *INO1* promoter

We have shown that *INO1* expression and the recruitment of chromatin remodelers are not influenced in the absence of Snf2p acetylation. Furthermore, we also observed the accumulation of both remodelers at the *INO1* promoter. However, those chromatin remodelers might be required for the expression of other genes within yeast cells. Therefore, the accumulation of Ino80p and Snf2p at the *INO1* promoter might impact their availability for other genes.

   To demonstrate whether this limited availability of remodelers in unAcSnf2p cells can influence their regulated pathways, plate sensitivity assays were performed targeting osmoregulation [21], DNA damage [22], carbon source usage [23], and copper toxicity [20]. WT, unAcSnf2p, and *snf2Δ* strains were first plated on SC media to demonstrate cell viability. All strains demonstrated viable growth at the five dilutions used (Fig 5A; stock cell concentration $1.6x10^9$ cells per ml diluted $10^{-1}$ through $10^{-5}$). These same strains were then analyzed under high osmolarity conditions. Osmoregulation is a common stress pathway that is highly regulated in yeast and can easily be tested for dysregulation via plate sensitivity assays, in which yeast growth is screened in the presence of high salt concentrations [24]. In the presence of high salt (0.8 M KCl), the WT strain was still visible at the greatest dilution (Fig 5B). The unAcSnf2p strain demonstrated growth comparable to the WT at most dilutions, but was not observed at the greatest dilution of $10^{-5}$. Lastly, the *snf2Δ* exhibited significantly diminished

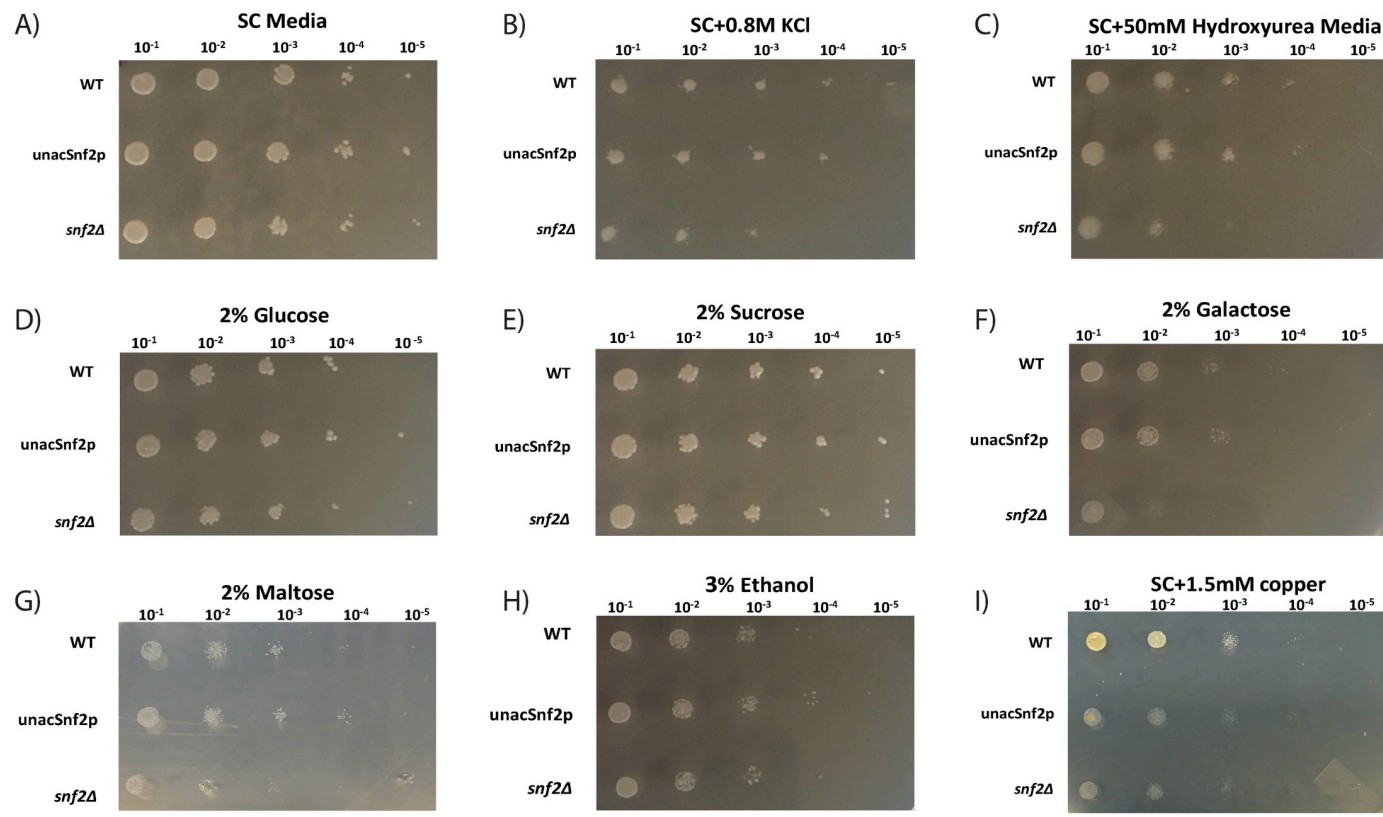

**Fig 5. Increased copper sensitivity when Snf2p is unacetylatable.** WT, unAcSnf2p, and *snf2Δ* cellular growth in (A) SC media, (B) SC with 0.8M KCl, (C) SC with 50mM Hydroxyurea (D) 2% Glucose, (E) 2% Sucrose, (F) 2% Galactose, (G) 2% Maltose, (H) 3% Ethanol, and (I) 1.5mM copper. All cells were grown overnight, and then diluted to a starting optical density of 2.0 at 600nm absorbance as a stock solution (approximately $1.6x10^9$ cells per ml). Five serial dilutions were plated for each strain (from left to right $10^{-1}$, $10^{-2}$, $10^{-3}$, $10^{-4}$, and $10^{-5}$). Experiments were repeated in triplicate.

growth compared to the other strains, as it had very weak growth at the $10^{-3}$ dilution and no growth at the $10^{-4}$ or $10^{-5}$ dilutions. As the unAcSnf2p strain demonstrated growth more comparable to the WT versus the deletion strain, the sensitivity of the cells in the presence of osmotic stress did not appear to be significantly altered in the absence of Snf2p acetylation.

Similar results were observed with regard to DNA damage sensitivity in which all strains were plated in the presence of 50 mM hydroxyurea. In the presence of hydroxyurea, the WT strain and the unAcSnf2p strain both demonstrated similar growth with each being observed through the $10^{-4}$ dilution, whereas the *snf2Δ* strain was unable to viably grow beyond the $10^{-2}$ dilution (Fig 5C). As with the osmotic results, the DNA damage sensitivity plates demonstrated that the lack of Snf2p acetylation did not hinder DNA damage repair.

Another highly regulated process in yeast cells, which is known to be associated with SWI/SNF family remodelers, is the utilization of various carbon sources in the form of fermentable substrates, such as glucose, sucrose, galactose, and maltose, as well as non-fermentable carbon sources, such as ethanol [23–25, 26]. In *snf2Δ* strains, for example, this regulatory mechanism is known to be disrupted so cells grow slower in sucrose media, and fail to grow in galactose [25]. To determine if the availability of Snf2p affects the utilization of alternate carbon sources, synthetic complete plates with 2% glucose, 2% sucrose, 2% galactose, 2% maltose, or 3% ethanol were utilized. We observed that the unAcSnf2p strain survived all conditions, including the fermentable carbon sources, such as 2% glucose, 2% sucrose, 2% galactose, and 2% maltose, as well as the non-fermentable carbon source of 3% ethanol (Fig 5D, 5E, 5F, 5G and 5H). In fact, on each plate, the unAcSnf2p mutant grew comparable to the WT. These results suggest that the accumulation of both Ino80p and Snf2p at the *INO1* promoter does not affect the utilization of alternate carbon sources.

Recent research has demonstrated that the chromatin remodelers, Snf2p and Ino80p are also required at the *CUP1* promoter [20]. Thus, to determine if Snf2p and Ino80p accumulation at the *INO1* promoter is detrimental to copper resistance, all strains were plated on synthetic complete plates containing 1.5 mM. In the presence of 1.5 mM copper, WT cells demonstrated strong viability until the $10^{-4}$ dilution, whereas the unAcSnf2p strain demonstrated significant sensitivity, with very weak growth even at the $10^{-1}$ dilution, comparable to the *snf2Δ* strain (Fig 5I). This suggests that Snf2p and Ino80p accumulation at the *INO1* promoter has dramatic influence on yeast cells' resistance to copper.

## The increased copper sensitivity resulted from the decreased *CUP1* expression

To further examine the detail of increased copper sensitivity for unAcSnf2p strain, WT cells, unAcSnf2p, and *snf2Δ* cells were grown in the presence or absence of copper under un-induced (100 μM inositol) and induced conditions (0 μM inositol) for *INO1*.

In 0 μM inositol, all three strains were able to grow in the absence of copper as expected, although *snf2Δ* cells exhibited a significantly reduced growth pattern compared to when inositol was present. As expected, there was no significant difference between WT and unAcSnf2p growth in these circumstances when no toxic copper was present. In the presence of 1.5 mM copper and 0 μM inositol, however, the WT cells were once again able to survive, while the unAcSnf2p cells were now sensitive to the copper, comparable to *snf2Δ* cells (Fig 6A). The unAcSnf2p cells only reached an $A_{600nm}$ of 0.59 after 24 hours. This is when chromatin remodelers have dissociated from the *INO1* promoter in WT cells but remain accumulated in unAcSnf2p cells. This suggests that copper toxicity defense is hindered in unAcSnf2p cells under *INO1* inducing conditions when chromatin remodelers are accumulated at the *INO1* promoter.

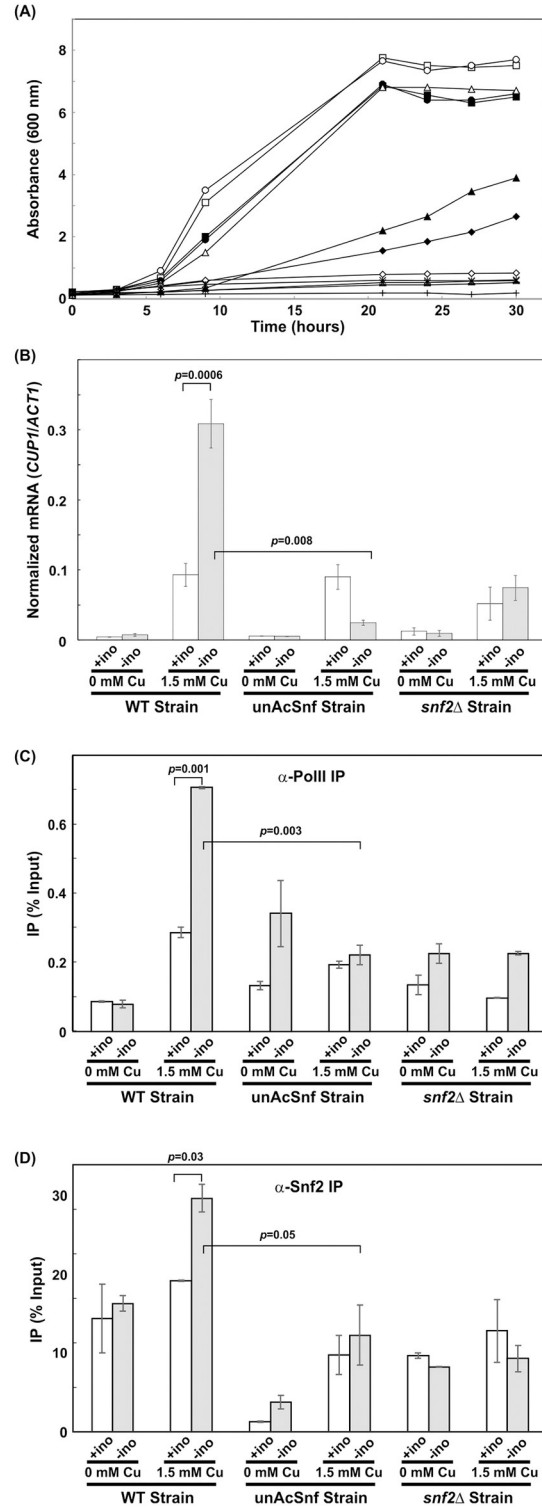

**Fig 6. Snf2p acetylation is required for copper toxicity survival.** (A) Growth of yeast cells in the presence or absence of 100 μM myo-inositol with or without 1.5 mM copper. WT cell: (■) SC—inositol and 0 mM $Cu^{2+}$, (▫) SC + inositol and 0 mM $Cu^{2+}$, (♦) SC—inositol and 1.5 mM $Cu^{2+}$, (◇) SC + inositol and 1.5 mM $Cu^{2+}$. Mutant UnAcSnf2p cell: (●) SC—inositol and 0 mM $Cu^{2+}$, (○) SC + inositol and 0 mM $Cu^{2+}$, (*) SC—inositol and 1.5 mM $Cu^{2+}$, (x) SC + inositol and 1.5 mM $Cu^{2+}$. Mutant *snf2*Δ cell: (▲) SC—inositol and 0 mM $Cu^{2+}$, (Δ) SC + inositol and 0 mM $Cu^{2+}$, (+) SC—inositol and 1.5 mM $Cu^{2+}$, (-) SC + inositol and 1.5 mM $Cu^{2+}$. Cells were grown in SC media and were subsequently

washed and diluted to an A$_{600nm}$ of 0.2 in repressing or inducing conditions, 100μM inositol (+ino) and 0μM inositol (-ino) synthetic complete media, respectively, with or without 1.5 mM copper. (B) *INO1* mRNA was examined by qRT-PCR analysis. Cells were grown in SC media to mid-log phase (A$_{600}$ = 1.0) and were subsequently washed and subjected to *INO1* repressing (100 μM inositol) or inducing conditions (0 μM inositol), as well as *CUP1* repressing (0 mM copper) and inducing conditions (1.5 mM copper), respectively. Cells were grown for 2 hours prior to collection. The expression ratio for *INO1* mRNA/*ACT11* mRNA is graphed as mean standard ± deviation. All experiments were repeated at least three times, and in each experiment PCR reactions were done in triplicate. (C-D) Real-time PCR analysis of DNA immunoprecipitated through ChIP with an antibody against (C) RNA polymerase II (α-Pol II) and (D) FLAG-tagged Snf2p (α-Snf2p) at the *CUP1 TATA* or *UAS* promoter, respectively, for WT cells, unAcSnf2p, and *snf2Δ* cells that were grown to mid-log phase (A$_{600}$ = 1.0) in SC, and were subsequently washed and subjected to *INO1* repressing (100 μM inositol) or inducing conditions (0 μM inositol), as well as *CUP1* repressing (0 mM copper) and inducing conditions (1.5 mM copper), respectively. Cells were grown for 2 hours prior to collection. All experiments were performed with two repeats and all PCR reactions were done in at least triplicate. The IP for the *CUP1 TATA* or *UAS* promoter is graphed as mean ± standard deviation normalized to input and mock.

We then examined cell growth in 100 μM inositol, where cells are known to have Snf2p highly recruited and accumulated at the *INO1* promoter [11]. In the absence of toxic copper, all three strains, including the *snf2Δ* cells, demonstrated strong growth in 100 μM inositol with no significant difference observed between the WT and unAcSnf2p cells. Unlike the 0μM inositol media, however, WT cells in 100μM inositol, when chromatin remodelers are most heavily accumulated at the *INO1* promoter, were no longer able to survive in the presence of 1.5 mM copper and now demonstrated a lack of growth comparable to the unAcSnf2p cells and the *snf2Δ* cells (Fig 6A). As such, in the presence of 1.5mM copper which is *CUP1* inducing conditions, there was a significant difference between 0 μM inositol (when remodelers have dissociated away from the *INO1* promoter) and 100 μM inositol WT growth (when remodelers are most heavily accumulated at the *INO1* promoter). On the other hand, the unAcSnf2p strain demonstrated growth defect in the presence of copper under both 0 μM and 100 μM inositol conditions. As such, this suggests that *CUP1* expression is influenced when remodelers are accumulated at the *INO1* promoter.

To further examine whether *CUP1* expression is influenced by the accumulation of unacetylatable Snf2p at *INO1* promoter, *CUP1* mRNA were analyzed. In the absence of copper, *CUP1* is not expressed in each of the strains and inositol conditions (Fig 6B). In the presence of 1.5 mM copper, however, we observed a significant difference in *CUP1* mRNA levels between *INO1* repressing and inducing conditions in the WT cells. The WT *CUP1* mRNA levels are 0.093±0.016 and 0.309±0.035, under *INO1* repressing and inducing conditions, respectively. On the other hand, we observed no significant difference in *CUP1* mRNA levels for the unAcSnf2p strain under *INO1* repressing and inducing conditions. Similarly, we also found no significant difference in *CUP1* mRNA levels for the *snf2Δ* strain under *INO1* repressing and inducing conditions (Fig 6B). To rule out the possibility that it is not the stability of *CUP1* mRNA but instead transcription that is adversely affected in the unAcSnf2p mutant, we also examined RNA polymerase II recruitment at the *TATA* region of *CUP1* promoter [20]. We found that Pol II-IP values showed a similar pattern as the mRNA levels (Fig 6C). The only significant change was the increase of WT Pol II-IP values from repressing *CUP1* with 0.08% ±0.01% to inducing *CUP1* with 0.71%±0.003% when *INO1* was in inducing conditions ($p$ = 0.0004). All other conditions showed no significant difference. Therefore, these observations indicate that the lack of *CUP1* expression in unAcSnf2p strain under both *INO1* and *CUP1* inducing conditions is due to the absence of RNA polymerase II instead of the *CUP1* mRNA instability.

Previously, we have shown that Snf2p is present at the *CUP1* promoter and it is responsible for recruiting chromatin remodeling activity to the promoter for *CUP1* induction under induced conditions [20]. Here, we observed the decreased *CUP1* expression in unAcSnf2p

strain under *INO1* inducing conditions, suggesting that the unacetylated Snf2p is not recruited to *CUP1* promoter for induction. Therefore, it was instructed to examine whether the accumulation of Snf2p at unAcSnf2p strain's *INO1* promoter resulted in the lack of Snf2p at the *CUP1* promoter for *CUP1* induction.

For ChIP analysis, the Snf2-FLAG IP values were examined at the *CUP1* upstream activation sequences (*UAS*) [20]. Under *INO1* repressing conditions (100 μM inositol), the WT strain's Snf2-FLAG IP values demonstrated an insignificant difference ($p = 0.39$), as the Snf2p-FLAG-IP were 0.26%±0.08% under *CUP1* repressing conditions and 0.34% ±0.002% under *CUP1* inducing conditions, which demonstrated the accumulation of Snf2p at the *INO1* promoter but not the *CUP1* promoter region (Fig 6D). Under the same *INO1* repressing conditions, the Snf2p-FLAG-IP values in the unAcSnf2p mutant were 0.02% ±0.001% and 0.17% ±0.04% under *CUP1* repressing and inducing conditions, respectively (Fig 6D). The difference is insignificant ($p = 0.08$), and this observation indicated that unacetylated Snf2p does not accumulate at the *CUP1* promoter. The negative control, *snf2Δ*, demonstrated Snf2p-FLAG-IP values with no significant difference between repressing and inducing conditions ($p = 0.92$), with 0.17% ±0.01% and 0.23% ±0.07% under *CUP1* repressing and inducing conditions, respectively. These results suggested that Snf2p accumulates at the *INO1* promoter not at the *CUP1* promoter under *INO1* repressing conditions. Therefore, *CUP1* is not expressed even in the *CUP1* inducing conditions. This is in agreement with the results from growth experiments and RNA analysis (Fig 6A and 6B).

Under *INO1* inducing conditions (0 μM inositol), the WT strain's Snf2p-IP values were 0.29%±0.02% under *CUP1* repressing conditions and 0.52% ±0.03% under *CUP1* inducing conditions. This showed a significant difference ($p = 0.02$) and indicated the accumulation of Snf2p at the *CUP1* promoter region (Fig 6D). Under the same *INO1* inducing conditions, the Snf2p-IP values of the unAcSnf2p mutant were 0.07% ±0.02% and 0.22% ±0.07% under *CUP1* repressing and inducing conditions, respectively (Fig 6D). The difference is insignificant ($p = 0.16$), and this observation indicated that unacetylated Snf2p does not accumulate at the *CUP1* promoter. Again, *snf2Δ* strain's IP values showed no significant difference between repressing and inducing conditions ($p = 0.59$), with 0.15% ±0.001% and 0.17% ±0.03% under *CUP1* repressing and inducing conditions, respectively. These results suggested that Snf2p departs from the WT strain's *INO1* promoter and it accumulates at the *CUP1* promoter when both *INO1* and *CUP1* are induced. On the other hand, Snf2p accumulates at the unAcSnf2p strain's *INO1* promoter not at the *CUP1* promoter when both *INO1* and *CUP1* are induced. Therefore, Snf2p acetylation is important for its dissociation from the *INO1* promoter and to be recruited to the *CUP1* promoter when both are induced. Taken together with the growth analyses, the mRNA analyses and the RNA polymerase II IP analysis, the Snf2p-IP analysis confirmed that the accumulation of unacetylated Snf2p at the *INO1* promoter was detrimental to *CUP1* induction and proper copper protection.

## Discussion

Transcriptional activation involves a series of activators and coactivators interactions with the upstream regulatory region of the genes, as well as interactions with each other and with transcription machinery during the process. Although recent research shows that some histone acetylases (e.g., Gcn5p) are capable of acetylating chromatin remodelers (e.g., SWI/SNF, RSC, or ISWI) [16–17, 27–28], the implications of this post-translational modification in the process of transcriptional activation has yet to be thoroughly examined.

Through ChIP and qPCR, we demonstrated that Snf2p acetylation is required for Snf2p dissociation from the *INO1* promoter upon induction (Fig 1A). Furthermore, a lack of Snf2p

acetylation resulted in an accumulation of Ino80p at the *INO1* promoter (Fig 1B). Our previous findings had demonstrated that not only does Ino80p arrive at the *INO1* promoter before Snf2p, but that this binding of Ino80p is necessary in order to subsequently recruit Snf2p to the promoter [10]. Since the INO80 complex binds to the promoter first and then SWI/SNF binds, it is likely that the removal of the Snf2p from the region is necessary for INO80 to be free to dissociate. Therefore, it is obvious that the majority of Ino80p remains trapped at the *INO1* promoter region when unacetylatable Snf2p accumulates at *INO1* promoter.

Previously, we observed that the chromatin remodeler, Snf2p, accumulated at the *INO1* promoter in the absence of Gcn5p [11]. Since Snf2p is known to be acetylated by Gcn5p [17], Snf2p remains unacetylated in the absence of Gcn5p. As a result, this is similar to the unacetylatable Snf2p mutant where Ino80p accumulated at the *INO1* promoter under inducing conditions (Fig 1B). Therefore, with the use of an unacetylatable Snf2p mutant, we now provide direct evidence that the dissociation was due to the acetylation of Snf2p, with a lack of acetylation resulting in an accumulation of the chromatin remodelers (Fig 1). Furthermore, Ino80p failed to dissociate from the *INO1* promoter upon induction, and instead remained occupying the region, perhaps trapped by the accumulated unacetylated Snf2p, which is part of a much bulkier SWI/SNF complex.

Based on the finding that Snf2p acetylation was required for dissociation from the *INO1* promoter, it was necessary to next explore the possible implications of this dissociation with regard to transcriptional activation and coactivator activities. Regarding transcriptional activation, growth survival and mRNA analyses confirmed that although the Snf2p acetylation was necessary for dissociation from the promoter, it was not necessary for *INO1* gene expression, as the unAcSnf2p strain was able to survive in the absence of inositol and expressed normal *INO1* mRNA levels under this inducing condition (Fig 2). This observation is in agreement with our previous finding that both Gcn5p and Esa1p are recruited to *INO1* promoter after the start of *INO1* induction [11]. Furthermore, our findings support the *INO1* induction model where remodelers are required to be acetylated for dissociating from the promoter after the induction [8, 10–11].

*INO1* activation is a highly regulated process that is heavily dependent upon the recruitment and dissociation of various coactivators, such as the chromatin remodelers, Snf2p and Ino80p, in order to modify the nucleosome structure at the promoter region and promote appropriate polymerase activity [8, 10–11]. Since we demonstrated that Snf2p acetylation was responsible for the dissociation of Snf2p from the *INO1* promoter and the accumulation of both Snf2p and Ino80p was not detrimental to *INO1* expression, it is reasonable to predict that chromatin remodeling activities and the recruitment of RNA polymerase II of unAcSnf2p strain should be the same as WT strain under inducing conditions. Indeed, we did observe that the nucleosome density and histone acetylation patterns are very similar for both WT and unAcSnf2p strains (Fig 3). Overall, our results suggested that acetylation plays a role beyond transcriptional activation and may instead have additional significance, possibly in the recycling of remodelers to mobilize elsewhere as proposed previously [11]. As a result, it is likely to affect the various actions that Ino80p/Snf2p would require to perform upon dissociation from the promoter. This is because both Ino80p and Snf2p are critical yeast chromatin remodelers responsible for regulating more than just the *INO1* gene. Although hydroxyurea-induced DNA damage repair, osmoregulation, and carbon source utilization were not significantly affected by the accumulation of Snf2p at the *INO1* promoter (Fig 5), protection from copper toxicity was noticeably impeded, as demonstrated by sensitivity plate assays (Fig 5I) and growth analyses (Fig 6A). These results suggested that Snf2p must depart from the *INO1* promoter in order to be readily available at the necessary levels for the induction of *CUP1*, which also requires both Ino80p and Snf2p before it is able to protect cells from copper via

metallothionine production [20]. This hypothesis was further supported by the observations that unAcSnf2p cells were shown to lack *CUP1* induction under conditions where Snf2p was accumulated at the *INO1* promoter (Fig 6B). Furthermore, we also observed that Snf2p were not recruited to *CUP1* promoter and this resulted in the *CUP1* repression (Fig 6D). Therefore, our findings demonstrated that Snf2p should be recycled for other purpose. Perhaps the recycle of remodelers is an economical strategy in the cells to conserve the cell metabolism.

The order of events for *INO1* and *CUP1* induction has been examined and they demonstrate inverse patterns of remodeler recruitment [10, 20]. Under both *INO1* and *CUP1* inducing conditions, Snf2p is required to be recruited to the *CUP1* promoter but the Snf2p departs from the *INO1* promoter. Therefore, if the Snf2p is trapped at the *INO1* promoter, less Snf2p will be available for *CUP1* activation in the presence of copper. Here, we revealed that when chromatin remodelers are trapped at the *INO1* promoter, the copper toxicity defense pathway, which is dependent upon those same remodelers for activation of *CUP1*, is unable to properly function and protect the cells. Our results suggest that acetylation of remodelers is necessary to allow the recycle of remodelers and a working model has been proposed according to this finding (Fig 7). In this model, the acetylation status of remodelers determines whether remodelers can bind to promoter region or not. We propose that both unacetylated Ino80p and Snf2p bind to the *INO1* promoter before induction. Once *INO1* is induced, histone acetylases Gcn5p and Esa1p acetylate chromatin remodelers Snf2p and Ino80p, allowing the remodelers to dissociate from the *INO1* promoter. Once freed from the *INO1* promoter region, the remodelers can be recruited to the *CUP1* promoter, which is the gene coding for the production of a metallothionein that binds free copper to protect yeast cells from toxicity, and activate *CUP1* expression. In the event that remodeler acetylation is absent, then remodelers are unable to vacate the *INO1* promoter and thus fail to mobilize towards *CUP1*. This yields major defects in copper detoxification and protection. As such, this model describes the strategy cells adopted to use remodelers efficiently in activating gene expression, and maximizing resources.

Our findings have revealed insight into the mechanism of chromatin remodeler acetylation and its implications in gene expression regulation. Chromatin remodeler mobility is an active area of research, with significant advances recently being made with regard to the RSC complex, for which there are only between 200 to 2000 copies regulating the nearly 70,000 nucleosomes in need of condensing within yeast [29]. This disparity between copy numbers of nucleosomes and the remodelers that regulate them is even more pronounced with regard to Snf2p with only 100–500 copies per nucleus [30], yet no studies have elucidated the implications at this point. So even though Snf2p acetylation was not necessary for *INO1* transcriptional activation, we demonstrated that it is still a significant post-translational modification in yeast cells, as our results strongly suggest a recycling role for increased mobility of remodelers may be in effect for chromatin remodeler acetylation. As Snf2p and Ino80p, known regulators of *CUP1*, accumulate at the *INO1* promoter in the absence of Snf2p acetylation, other genes requiring Snf2p and Ino80p may also be affected, since Snf2p is required for the activation of nearly 5% of all yeast genes, yet is fairly rare within the cells at an estimated 100–500 copies per nucleus [30]. Perhaps, the inverse remodeler recruitment pattern observed with certain genes, such as *INO1* compared to *CUP1*, is normal in the cells and may provide maximum and efficient usage of the remodelers for the transcriptional activation.

## Materials and methods

### Yeast strains and growth conditions

Wild Type (BY4741; *MATa his3Δ200 leu2Δ0 met15Δ0 ura3Δ0 SNF2-FLAG::LEU*; a gift from Dr. Workman's lab [17]) and unAcSnf2p (*MATa his3Δ200 leu2Δ0 met15Δ0 ura3Δ0*

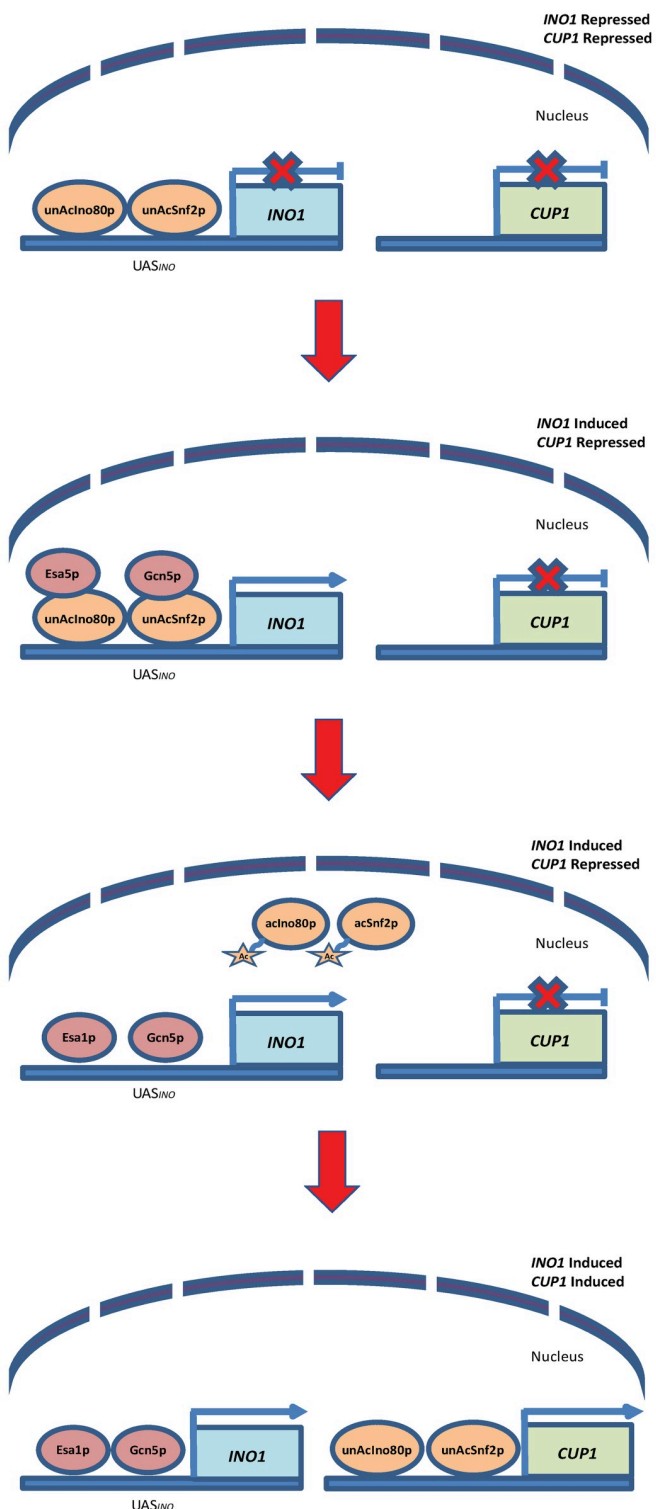

**Fig 7. Proposed model of chromatin remodeler's recycling during gene activation.** In this model, both unacetylated Ino80p and Snf2p bind to the *INO1* promoter before induction. Once *INO1* is induced, histone acetylases Gcn5p and Esa1p acetylate chromatin remodelers Snf2p and Ino80p, allowing the remodelers to dissociate from the *INO1* promoter. Once freed from the *INO1* promoter region, the remodelers can be recruited to the *CUP1* promoter and activate *CUP1* expression.

*snf2-K1493R-K1497R-FLAG::LEU*; a gift from Dr. Workman's lab [17]), as well as *snf2Δ* (*MATa his3Δ200 leu2Δ0 met15Δ0 ura3Δ0 snf2Δ*) strains were used in this study. WT and unAcSnf2p cells were grown at 30˚C in Synthetic Complete (SC) media without leucine (SC-leu) containing 2% glucose (wt/vol). *Snf2Δ* cells were grown at 30˚C in SC. Upon optical density (O.D.) between 0.8–1.2, cells were centrifuged and washed twice with media lacking any inositol to completely remove inositol. Cells were then resuspended in appropriate media with 100 μM inositol for repressing conditions or without inositol for inducing conditions, and incubated for 2 hours at 30˚C.

## Chromatin immunoprecipitation (ChIP)

Yeast strains were grown to mid-logarithmic phase (0.8 $A_{600nm}$) in 10 μM inositol synthetic complete (SC) media, washed and subjected to conditions to repress (100 μM inositol) or induce (0 μM inositol) *INO1* transcription for 2 hours prior to collection. For switching *INO1* expression conditions experiment, cells were subjected to the inducing conditions (0 μM inositol) for 1 hour followed by the repressing conditions (100 μM inositol). For copper experiment, cells were grown initially in 10 μM inositol SC media and subsequently washed and diluted to an $A_{600nm}$ of 0.2 in *INO1* repressing or inducing conditions as before, that also contained either 0 mM Cu or 1.5mM Cu. The preparation of cross-linked chromatin, immunoprecipitation procedures, and antibodies used in this study are described in the S1 Table.

Real-time qPCR using Taqman probes (Applied Biosystems) was employed to analyze IP samples, along with mock and input samples. All experiments were performed with three repeats and all PCR reactions were done in at least duplicate. The IP for each sample was calculated as:

$$\left[2^{-((\text{IP } Ct - \text{mock } Ct) - (\text{input } Ct - \text{mock } Ct))}\right] * (100/1100) * (A_{260} * 50\mu\text{g/ml} * 35 \text{ dilution factor})$$

The IP for the *INO1* or *CUP1* promoter was then graphed as mean standard ± deviation normalized to input and mock. Input was prepared as all genomic DNA sequences from the cell lysate without any selection or immunoprecipitation. Mock, on the other hand, was prepared as a no-antibody signal background in which all ChIP steps were performed on cell lysate, except for the addition of the selective antibody. Primer and Probe sequences utilized are listed in S2 and S3 Tables.

## Growth analysis

Cells were grown initially in 10 μM inositol SC media until mid-log phase and subsequently washed and diluted to an $A_{600nm}$ of 0.2 in repressing or inducing conditions as before. Cell growth was monitored spectrophotometrically using the $A_{600}$ density until stationary phase was reached. Repeat colonies were averaged and standard deviation was calculated.

## RNA preparation and qRT-PCR analysis

Cells were grown to mid-log phase (0.8 $A_{600nm}$) in 10 μM inositol SC media, washed and subjected to repressing (100 μM inositol for *INO1*; 0 mM Cu for *CUP1*) or inducing conditions (0 μM inositol for *INO1*; 1.5 mM Cu for *CUP1*) for 2 hours prior to analyses. For switching *INO1* expression conditions experiment, cells were subjected to the inducing conditions (0 μM inositol) for 30 min or 1 hour followed by the repressing condition (100 μM inositol) for 1 hour 30 min or 30 min, respectively. The total RNA preparation and qRT-PCR analysis was performed as described in the S1 File. The procedures of preparation and the analysis of target DNA sequence quantities are described in the S1 File as well, along with all real-time PCR

primers. All experiments were performed with three repeat colonies and all PCR reactions were performed in duplicate. RNA expression levels were normalized to the constitutive *ACT1* housekeeping gene using the formula $2^{-\Delta Ct}$ in which *ΔCt* represents the difference between the *Ct* value of *INO1* or *CUP1* and the *Ct* value of *ACT1*. Final data was graphed as mean ± standard deviation.

### Sensitivity assays

Cells were grown overnight in SC media at 300rpm, 30˚C to saturation and subsequently diluted with autoclaved water to an $A_{600nm}$ of 2.0 (approximately 1.6 x $10^9$ cells/ml). Five 10-fold serial dilutions of all strains were performed using autoclaved water. Each dilution was plated as a 2μl droplet onto SC plates as a control to demonstrate cell viability. Dilutions were then plated on SC plates containing 50 mM hydroxyurea, 0.8M KCl, 1.5mM copper and 3mM copper [20]. Lastly each serial dilution was plated on 2% Glucose plates, 2% Sucrose plates, 2% Galactose plates, 2% Maltose plates, and 3% Ethanol plates [23–25]. Plates were incubated for 48 hours at 30˚C. All experiments were repeated in duplicate.

### Copper sensitivity growth analyses

Cells were grown initially in 10 μM inositol SC media and subsequently washed and diluted to an $A_{600nm}$ of 0.2 in repressing or inducing conditions as before, that also contained either 0 mM Cu, 1.5mM Cu, or 3 mM Cu. Cell growth was monitored spectrophotometrically using the $A_{600nm}$ density until stationary phase was reached. All growth experiments were performed in duplicate; repeats were averaged and standard deviation was calculated.

## Supporting information

**S1 File.**
(DOCX)

**S1 Dataset.**
(XLSX)

**S1 Table. Antibodies used for ChIP experiments.**
(DOCX)

**S2 Table. Primer sequences for reverse transcriptase PCR.**
(DOCX)

**S3 Table. Primer and probe sequences for Real-time qPCR.**
(DOCX)

## Acknowledgments

We are grateful to our laboratory colleagues for technical assistance and comments on the manuscript.

## Author Contributions

**Data curation:** Michelle Esposito, Goldie Libby Sherr, Anthony Esposito, George Kaluski, Farris Ellington.

**Formal analysis:** Michelle Esposito, Goldie Libby Sherr, Chang-Hui Shen.

**Funding acquisition:** Chang-Hui Shen.

**Investigation:** Michelle Esposito, Goldie Libby Sherr, Anthony Esposito, George Kaluski, Farris Ellington.

**Methodology:** Michelle Esposito.

**Supervision:** Chang-Hui Shen.

**Writing – original draft:** Michelle Esposito, Anthony Esposito, Chang-Hui Shen.

**Writing – review & editing:** Chang-Hui Shen.

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
