## [Decision Letter · Decision Letter 0]

28 Oct 2019

PONE-D-19-27586

Accumulation of Unacetylatable Snf2p at the INO1 promoter is detrimental to remodeler recycling supply for CUP1 induction.

PLOS ONE

Dear Dr. Shen

Thank you for submitting your manuscript to PLOS ONE. After careful consideration, we feel that it has merit but does not fully meet PLOS ONE’s publication criteria as it currently stands. Therefore, we invite you to submit a revised version of the manuscript that addresses the points raised during the review process.

Both reviewers, who are expert in the field, found the study interesting and well-done. However they have raised some very pertinent questions, including that whether the reduced expression in the Snf2 mutant direct effect of diminished availability of Snf2 at CUP1 promoters. Given the importance of the findings reported in this paper, it is important to address the concerns raised by the reviewers.

We would appreciate receiving your revised manuscript by December 24, 2019. To enhance the reproducibility of your results, we recommend that if applicable you deposit your laboratory protocols in protocols.io, where a protocol can be assigned its own identifier (DOI) such that it can be cited independently in the future. For instructions see: http://journals.plos.org/plosone/s/submission-guidelines#loc-laboratory-protocols

We look forward to receiving your revised manuscript.

Kind regards,

Chhabi K. Govind, Ph. D.

Academic Editor

PLOS ONE

Journal Requirements:

2) Please amend the subsection category “[FOR JOURNAL STAFF USE ONLY]” for your manuscript. Unfortunately, this is not a valid category. At this time, please choose one or more subsections that best represent the topic(s) of your study.

3) Thank you for stating the following in the Acknowledgments Section of your manuscript:

[This work was supported by an NATO Grant (NATO SPS G5266), an NSF Grant

(MCB-0919218) and PSC-CUNY awards to C.-H. S.]

 [CHS - NATO SPS G5266 and NSF MCB 0919218

NO - Include this sentence at the end of your statement: The funders had no role in

study design, data collection and analysis, decision to publish, or preparation of the

manuscript.]

Please include the updated Funding Statement in your cover letter. We will change the online submission form on your behalf.

4) We note that you have included the phrase “data not shown” in your manuscript. Unfortunately, this does not meet our data sharing requirements. PLOS does not permit references to inaccessible data. We require that authors provide all relevant data within the paper, Supporting Information files, or in an acceptable, public repository. Please add a citation to support this phrase or upload the data that corresponds with these findings to a stable repository (such as Figshare or Dryad) and provide and URLs, DOIs, or accession numbers that may be used to access these data. Or, if the data are not a core part of the research being presented in your study, we ask that you remove the phrase that refers to these data.

5)  In your Data Availability statement, you have not specified where the minimal data set underlying the results described in your manuscript can be found. PLOS defines a study's minimal data set as the underlying data used to reach the conclusions drawn in the manuscript and any additional data required to replicate the reported study findings in their entirety. All PLOS journals require that the minimal data set be made fully available. For more information about our data policy, please see http://journals.plos.org/plosone/s/data-availability.

6) Please include captions for your Supporting Information files at the end of your manuscript, and update any in-text citations to match accordingly. Please see our Supporting Information guidelines for more information: http://journals.plos.org/plosone/s/supporting-information.

Reviewers' comments:

Reviewer's Responses to Questions

**Comments to the Author**

1. Is the manuscript technically sound, and do the data support the conclusions?

Reviewer #1: Yes

Reviewer #2: Yes

2. Has the statistical analysis been performed appropriately and rigorously? 

Reviewer #1: Yes

Reviewer #2: Yes

3. Have the authors made all data underlying the findings in their manuscript fully available?

Reviewer #1: Yes

Reviewer #2: Yes

4. Is the manuscript presented in an intelligible fashion and written in standard English?

Reviewer #1: Yes

Reviewer #2: Yes

5. Review Comments to the Author

Reviewer #1: INO1 is a well-known gene for studying transcriptional regulation in budding yeast. A number of transcription factors have been shown to influence expression of INO1. The focus of attention of this investigation is the coactivator Snf2. Authors demonstrated that the release of Ino80 and Snf2 from the INO1 promoter upon induction is regulated by acetylation of Snf2. In the unacetylable mutant of Snf2, neither Ino80 nor Snf2 are released from the INO1 promoter under inducive condition. The persistent occupancy of Ino80 and Snf2, however, has absolutely no adverse effect on the chromatin structure and transcription of INO1. Authors proposed that the amount of Snf2 in yeast cells is limiting, and therefore in the absence of release of Snf2 from INO1 in the mutant, transcription of other Snf2-requiring genes may be affected. They corroborated their hypothesis by demonstrating that the mRNA level of CUP1 is adversely affected in unacetylable mutant of Snf2 in the presence of copper and absence of inositol. These findings are significant and make an interesting scientific story. The manuscript can be improved further if authors can take care of the following issues:

(1) Authors have demonstrated that the steady state level of CUP1 mRNA decreases in the unacetylable mutant of Snf2 upon copper induction in the absence of inositol. To rule out the possibility that it is not the stability of CUP1 mRNA but transcription that is adversely affected in Snf2 mutant, authors need to perform either strand-specific TRO (Transcription Run-On) assay or RNAPII-ChIP as shown in Fig 3E.

(2) To complete the story, authors need to demonstrate by Snf2-ChIP that the unacetylable mutant of Snf2 is not recruited to the CUP1 promoter upon copper induction under inositol depleted conditions.

(3) The discussion is too long and at places redundant with Results. A shorter and focused ‘Discussion’ will improve the impact of the paper.

Reviewer #2: The article by Shen et al, is an elegant study on the role of Snf2 acetylation in regulating expression of genes. It carries forward studies from the Workman lab and provides a further detailed basis for how acetylation of Snf2 can affect gene expression by regulating other chromatin remodelers and suppress gene expression. The authors show, as would be predicted, that loss of Snf2 acetylation delays release of Snf2 from gene promoters. Very interestingly they show that Snf2 acetylation also affects release of INO80 from gene promoters. Additionally, the authors show that failure to release and recycle Snf2, decreases CUP1 expression. This is a well done study and should be accepted for publication after the few comments below are addressed.

Specific comments:

1) Can the authors show that loss of CUP1 expression in the unacetylable Snf2 is due to failure of Snf2 to be recruited to CUP1 promoter by ChIP?

2) Loss of Snf2 acetylation primarily affects release of Snf2. If the authors first induce INO1 and then suppress INO1 by adding back Inositol, does this affect Snf2, Ino80, PolII occupancy at INO1 and affect gene expression and histone levels?

6. PLOS authors have the option to publish the peer review history of their article (what does this mean?). If published, this will include your full peer review and any attached files.

Reviewer #1: No

Reviewer #2: No

---

## [Author Response · Author response to Decision Letter 0]

15 Feb 2020

Response to Reviewers:

Reviewer #1:

Point#1

To rule out the possibility that low CUP1 mRNA levels is not the stability of CUP1 mRNA but instead transcription that is adversely affected in the unAcSnf2p mutant, we have performed the RNA polymerase II-ChIP on CUP1 promoter region as directed by the reviewer. We have demonstrated that CUP1 expression in unAcSnf2p strain is significantly repressed under INO1 inducing conditions, and this is due to the inability to recruit RNA polymerase. The result has been shown in Figure 6C.

Point #2

To show that the loss of CUP1 expression in the unAcSnf2p strain (Snf2p is unacetylable in this strain) is due to the failure of Snf2p recruitment to CUP1 promoter, we have performed the Snf2p IP. Our results showed that Snf2p is not recruited to unAcSnf2 strain’s CUP1 promoter under INO1 inducing conditions. This confirmed that the unacetylable Snf2p accumulated at INO1 promoter under INO1 inducing conditions and this unacetylable Snf2p accumulation resulted in the failure of Snf2p recruitment at CUP1 promoter when CUP1 is under inducing conditions. The result has been shown in Figure 6D.

Point #3

We revised the Discussion section by removing some paragraphs that are redundant to the Result section, so that the revised manuscript is more concise and clear. Please see the marked-up copy for the detail.

Reviewer #2:

Point #1

This comment is exactly the same the Reviewer #1’s Point #2 comment and we have responded the comment. Please see the response (2) under Reviewer #1.

Point #2

Since the loss of Snf2p acetylation primarily affects the release of Snf2p, we also performed the switching INO1 expression conditions experiment as directed by the reviewer. Our results showed that INO1 expression, nucleosome density, recruitment of remodelers, and remodeling activities are independent of Snf2p acetylation when switching from inducing conditions to repressing conditions. The results has been presented in Figure 4.

---

## [Decision Letter · Decision Letter 1]

4 Mar 2020

Accumulation of Unacetylatable Snf2p at the INO1 promoter is detrimental to remodeler recycling supply for CUP1 induction.

PONE-D-19-27586R1

Dear Dr. Chang-Hui Shen,

We are pleased to inform you that your manuscript has been judged scientifically suitable for publication and will be formally accepted for publication once it complies with all outstanding technical requirements.

With kind regards,

Chhabi K. Govind, Ph. D.

Academic Editor

PLOS ONE

Additional Editor Comments (optional):

Reviewers' comments:

Reviewer's Responses to Questions

**Comments to the Author**

1. If the authors have adequately addressed your comments raised in a previous round of review and you feel that this manuscript is now acceptable for publication, you may indicate that here to bypass the “Comments to the Author” section, enter your conflict of interest statement in the “Confidential to Editor” section, and submit your "Accept" recommendation.

Reviewer #1: All comments have been addressed

Reviewer #2: All comments have been addressed

Reviewer #3: All comments have been addressed

2. Is the manuscript technically sound, and do the data support the conclusions?

Reviewer #1: Yes

Reviewer #2: Yes

Reviewer #3: Yes

3. Has the statistical analysis been performed appropriately and rigorously? 

Reviewer #1: Yes

Reviewer #2: Yes

Reviewer #3: Yes

4. Have the authors made all data underlying the findings in their manuscript fully available?

Reviewer #1: Yes

Reviewer #2: Yes

Reviewer #3: Yes

5. Is the manuscript presented in an intelligible fashion and written in standard English?

Reviewer #1: (No Response)

Reviewer #2: Yes

Reviewer #3: Yes

6. Review Comments to the Author

Reviewer #1: (No Response)

Reviewer #2: The authors have addressed the reviewers concerns to the reviewers satisfaction. The article should be accepted for publication.

Reviewer #3: (No Response)

7. PLOS authors have the option to publish the peer review history of their article (what does this mean?). If published, this will include your full peer review and any attached files.

Reviewer #1: No

Reviewer #2: No

Reviewer #3: No

---

## [Editor Report · Acceptance letter]

9 Mar 2020

PONE-D-19-27586R1 

Accumulation of Unacetylatable Snf2p at the <I>INO1</I> promoter is detrimental to remodeler recycling supply for <I>CUP1</I> induction. 

Dear Dr. Shen:

I am pleased to inform you that your manuscript has been deemed suitable for publication in PLOS ONE. Congratulations! Your manuscript is now with our production department. 

With kind regards,

on behalf of

Dr. Chhabi K. Govind 

Academic Editor

PLOS ONE